# Developing Public Health Promotion Strategies for Social Networking Sites: Perspectives of Young Immigrant Women in Norway

**DOI:** 10.3390/ijerph20054033

**Published:** 2023-02-24

**Authors:** Rita Agdal, Ingrid Onarheim Spjeldnaes

**Affiliations:** Faculty of Health and Social Sciences, Campus Bergen, Western Norway University of Applied Sciences, 5020 Bergen, Norway

**Keywords:** social networking sites, social media, digital health promotion, restrictive parenting, negative social control, settings-based health promotion

## Abstract

Background: Social networking sites (SNS) have emerged as digital settings for youth participation and health promotion. Understanding the complex dynamic of analog/digital participation has become crucial for settings-based health promotion strategies that aim to enable people to increase control over their health and environments. Previous research demonstrates that SNS influence young people’s health in complex ways, but less is known about how processes related to intersectionality are reflected in digital settings. This study asked the following question: how do young women with immigrant backgrounds experience and navigate SNS and how can this inform settings-based health promotion strategies? Methods: The study included three focus groups with 15 women aged 16–26 years and used thematic content analysis. Findings and conclusion: Young women with immigrant backgrounds reported that transnational networks provided a sense of belonging. However, their presence on SNS strengthened negative social control and had consequences for endeavors to connect with local peers in both digital and analog settings. Both challenges and resources were amplified. The participants reported that sharing strategies to navigate complex networks was useful; they emphasized the importance of anonymous chats, they shared health-related information with extended networks with lower e-literacy, and they saw opportunities for the cocreation of health promotion strategies.

## 1. Introduction

The settings-based approach in health promotion has traditionally focused on analog settings. The World Health Organization (WHO) has defined a setting as “the place or social context in which people engage in daily activities in which environmental, organizational, and personal factors interact to affect health and wellbeing” [1]. “Health is created and lived by people within the settings of their everyday life; where they learn, work, play, and love” [2]. Currently, both digital and analog arenas are settings for health promotion, as places for interventions and as arenas for capacity building, community action, and social participation [3]. Social participation is considered key for developing various kinds of participatory skills [4,5], and thus, social participation can be a fundament for social engagement and empowerment [5] to achieve community action and capacity building. Different aspects of youth participation have increasingly been emphasized both in research and in reports related to health, community work, and planning, as well as in policy documents by international organizations such as the World Health Organization and the United Nations. A document from the UN argues that “*Through active participation, young people are empowered to play a vital role in their own development as well as in that of their communities, helping them to learn vital life skills, develop knowledge on human rights and citizenship and to promote positive civic action*” [6].

Digital settings such as social networking sites (SNS) have recently been targeted for health promotion [3]. SNS are web-based services where users interact, share, and connect with other users, with (semi)public profiles that they construct themselves [7]. The SNS have different designs and features that are changed over time, and most users relate to several digital arenas. Scholars have described SNS as promoting a participatory culture, where youth develop their voices and identities as media creators through ongoing interaction with peers and wide audiences [8]. Thus, it has been argued that SNS have emerged as settings for youth’s everyday life, and thus for youth participation, and for settings-based health promotion [3,9,10,11]. Participation in the digital social settings of SNS influences the health of young people in complex ways [12,13,14]. Consequently, digital arenas such as SNS have been identified as central social arenas for developing health promotion strategies [3,9,10,12,15,16,17].

Some studies indicate that SNS are a part of civil society that may promote increased participation, for instance, by providing platforms where young people express concerns about political and societal issues [18]. Nevertheless, there are strong indications that analog social hierarchies are reproduced through SNS [7,19]. Studies on marginalized youth in disadvantaged neighborhoods found that SNS contribute to the reproduction and amplification of negative social interactions [19]. SNS constitute a hypothetical public space, where participants develop user norms specific to digital space that represent a continuation and transcending of power structures in the analog world [20]. Participation in digital settings is enabled by digital agency, which is unequally distributed and of varying relevance in analog settings [20]. The reproduction of hierarchies in digital settings implies unequal possibilities for participation and exclusion, but how this relate to the widespread phenomenon of fear of missing out (FOMO) and actually missing out on social opportunities and the pursuit of belonging and popularity, e.g., [21,22], is unclear. The research remains inconclusive on the relationship between the increased use of SNS and experiences of wellbeing, loneliness, and mental health problems [13,14]. There are strong indications that *passive* use of SNS decreases wellbeing, with different findings for *active* use of SNS [23]. Participation on SNS has been related to optimism and creativity [15], as well as possible effects on health. The concept of youth participation in studies of SNS is often unspecified. 

More research is needed to identify causal relations between specific forms of SNS use and participation in digital settings, their impact on health and wellbeing [13,23,24], and their potential impact on participation in analog settings. The knowledge about processes of reproduction of social inequality through the use of SNS is also insufficient, although some differences related to intersectionality have been described [19,24,25,26]. There is a paucity of consideration of settings-based health promotion in digital settings, and few studies describe how SNS can be approached or employed in settings-based health promotion [3]. Some studies on digital health promotion interventions on SNS find that this setting could provide networks and social support, which influence youth’s social norms and health-related behavior [3]. How SNS influence health, and the benefits and challenges of participation on SNS, differ substantially within the youth population. Reviews stress the need to explore specific youth segments [3,13] and the dynamics of digital settings relevant to health promotion interventions for specific target groups [9,13], particularly disadvantaged groups [12]. Immigrant youth constitute one such segment, although factors such as varying reasons for immigration induce group diversity and influence health expectancies [27] and, subsequently, the need for targeted health promotion strategies. 

Research on young people with immigrant backgrounds in Norway reports that SNS participation supports the formation of cultural identities and connectedness to the place of ancestral origin [28,29] and provides opportunities to communicate according to their preferences [28,29,30]. SNS represent discursive arenas where immigrant youth create functional identities through blogs and video-sharing services [30]. These youths demonstrate increased diversity in these arenas compared to that presented in the public representation of immigrant youth via news channels [30]. An ethnographic study from Denmark reports that SNS might serve as an arena and a tool for young Muslim women to cautiously explore social participation beyond the watchful eyes of their diaspora and neighborhood [13,26]. Young women face greater challenges in SNS-related activities than young men [13,31], but intersectionality has hardly been considered. How young women with immigrant backgrounds experience SNS participation remains understudied and undertheorized, and the implications for the design of settings-based health promotion strategies have not been given sufficient consideration. Thus, this study aims to increase the understanding of participation dynamics for young women with immigrant backgrounds and to explore their lived experiences of participation. Accordingly, this study addresses the question of how young women with immigrant backgrounds experience and navigate SNS and how this may inform settings-based health promotion.

## 2. Theoretical Perspectives

Th cultural phenomenological perspective in social science research on youth is the theoretical foundation of this study. This perspective is in line with the emphasis on the social and cultural context as a premise for the settings-based approach in health promotion [3] and a context-based understanding of youth participation [5]. Instead of focusing on how youth adapts to existing norms and adult life, emphasis social participation and the risk of potential exclusion [32]. This implies a focus on participation in social milieus with peers, which is considered a foundation for social and emotional support and for developing competencies for participation needed in social settings with increasing complexity [5,32]. Recently, it has been argued that social exclusion, which is commonly perceived as inhibiting participation, in some circumstances also triggers social engagement in both analog [5] and digital settings [20]. The emphasis on youth agency has led to findings that “negotiating of hierarchies and inequalities online may be an empowering experience”; see [20] (p.23). The emphasis on agency is compatible with critical perspectives on SNS participation that acknowledge power relations in digital settings, including the intersectionality of class, gender, and race [19,20,33,34]. The focus on agency also implies lived experiences in everyday life, where the identification of specific experiences of exclusion and inclusion enables an understanding of strategies for participation [26,35]. Participation and everyday life relate to the description of how immigrant youth appropriate and navigate digital spaces and counter exclusion [20].

Other studies find that education, social trust, and local connections increase the participation of young people at risk of experiencing social exclusion [36]. Scholars suggest that different kinds of participation connect and that social participation forms an aspect of participation that may evolve in various contexts influencing youths’ agency, including. their ability to engage, both in everyday life and as helpers or activists [5]. It is well established that social participation and local connections in the community potentiallyincrease the youth participatation in social activism or engage in other settings [37]. It has been discussed whether the development of personal participatory skills in local settings works as a fundament for participation in other contexts [37]. The context-based understanding of participation invites explorations of how different types of participation and participatory skills may evolve in digital and analog settings for youth in different social positions. 

## 3. Methods

This study is based on an explorative, qualitative design since cross-cultural perspectives in navigating SNS are relatively unexplored. This study takes an inductive, “bottom-up”, cultural phenomenological approach to establish knowledge based upon lived experiences as elaborated on by the subjects involved. Such explorative qualitative studies do not intend to establish generalizable findings, but can be used as a preparation to establish knowledge needed for further studies with mixed or quantitative methods in the next phase [38]. 

The 15 participants in the study were recruited through nongovernmental organizations operated by and for immigrant women and by a community center in Norway. Key persons in the organizations distributed invitations to women aged between 16 and 26 years who had lived in Norway for a minimum of two years and who had a good command of the Norwegian language. Except for three participants, the women originated from the Greater Middle East or the Horn of Africa. They had refugee statuses or had come to Norway to reunite with family. The names used in the article are fictive.

Data collection comprises of three focus group discussions [39] carried out from December 2019 to March 2020. Each session lasted around three hours and was conducted in Norwegian. The sessions were moderated to create an atmosphere of safety and trust, and participants were encouraged to share experiences related to SNS. Their role as experts on their situation were emphasized. We introduced the objective of the study, along with details on consent and anonymity, including their joint obligation to ensure the anonymity of their coparticipants. 

The semistructured interview guide (in Appendix A) included topics related to SNS: mapping the participants’ use of SNS; their self-presentations; their experiences; and evaluations of their navigations. 

The study authors are ethnic Norwegian women experienced in research on youth in Norway and in other countries. The interest in learning from the young women regarding their SNS use appears to have provided a welcome opportunity for these young women to share and reflect on their experiences.

The Norwegian agency for data protection adheres to the General Data Protection Regulation and has approved this study (NSD 529153). Verbal and written information to audio-record the discussions and store the data was provided prior to signing the consent form. The participants understood that they could withdraw at any time and that participation was voluntary. Furthermore, it was emphasized that the study authors would remain accessible for inquiries and support.

The collected audiotapes were transcribed to text by a professional shortly after the focus group sessions. Initially, the authors became familiar with the data by (re-)reading the transcripts. Notes were made of what emerged as important topics, which were shared and discussed among the authors, as the two researchers involved in the process of analysis. 

The structured analysis was continued, bearing initial impressions in mind, by using a two-phased thematic content analysis [39,40] (see an example of the process in Table 1). The first phase was guided by an open coding procedure with the intention to stay as close as possible to the discourses from the focus group material. One matrix was developed per focus group, using Word files that included columns of the original transcript, initial coding, and commentaries. The coding followed three principles developed by Owen (in [40]; (1) repetition of key words and phrases (e.g., “blocking contacts”); (2) recurrence of meaning through various concepts and expressions (e.g., parents’ interference with SNS activities); and (3) forcefulness of meanings expressed nonverbally (e.g., expressions of frustration in body language when talking about comments from people in transnational networks). The last principle was based on notes of nonverbal cues during data collection.

The study authors searched for patterns of what appeared meaningful to the study participants. For instance, focus group discussions often drifted towards the young migrant women’s challenges in navigating divergent social groups on SNS. At this point, the phenomenological experiences that appeared similar across participants became noticeable and an area of focus. 

Some commonly observed experiences relayed within the focus groups were initially conceptually interpreted in terms of the young women “connecting their rooms” or “closing their doors”. Eventually though, through discussions and by revisiting the original transcripts, this interpretation fell short, as the “opening and closing of doors” on SNS appear tightly knit with social positioning. The young women in this study were often not positioned to make such strategic, rational choices in their navigations on SNS. Thus, the authors’ understanding of the young women’s navigations was adjusted to interpretations related to the significance of power relations and of social positioning across diverse cultural and national groups. 

The second phase took the form of “closed coding” [40] (p. 98), aimed at linking the discourses from the focus groups (e.g., power relations and status-based hierarchies between youth groups and within diaspora) with public health perspectives. Critical questions were posed in relation to the data material, about power relations and what they contained, and how this function in the contexts of the study participants. After identifying sufficient repetition, recurrency, and forcefulness of status-based hierarchies, the analysis was advanced by thematizing *agency* and *participation* [20,32,41] from a health promotion perspective. Eventually, this exercise generated the following four themes (Table 2): navigating bonding with peer groups; navigating restrictive parenting; reciprocal transnational networks; and the becoming of women in globally interconnected settings.

Various measures were taken to consider the validity of the analysis and to scrutinize whether the themes accurately reflect the meanings in the data material [39]. First, internal coherence and consistency of the themes were assessed. The four themes did not overlap. All pertinent aspects of each theme were covered, and these centered around the core idea of each specific theme. The analysis was built on citations that were carefully selected to adequately represent the data material. Care was taken not to misrepresent data, for instance, by overemphasizing idiosyncratic, anecdotal instances, while also acknowledging that single experiences may contribute to a fuller explanation of a theme. Alternative explanations were not ruled out, as themes in qualitative research rarely appear uncontested or without contradictions. Finally, within the second phase of analysis wherein relevant broader perspectives were brought in, strong links between analytic claims and theoretical concepts were secured.

## 4. Results

The young women in this study participated on SNS through a variety of social media apps (Instagram, Snapchat, WhatsApp, Facebook, Pinterest, TikTok, YouTube, VSCO, Kik, Viber, Messenger, and LinkedIn). Like young women without migration backgrounds [42], the women in this study used these apps throughout their daily lives for various reasons, such as communicating, searching for knowledge (e.g., health information or advice on practical issues), organizing daily life (events in school or with friends), searching for entertainment and inspiration, developing professional networks, staying in touch with family and diaspora networks, and participating in networks with peers. However, their position as minoritized women within their diaspora and transnational networks influenced their SNS activities. Not dissimilar to their analog social settings, SNS could actualize their resources as well as their concerns. 

The analysis revealed four themes of participation on SNS: (1) *Navigating bonding with peer groups* was important but could not be viewed in isolation from their other social networks. Specifically, strategies for SNS participation were framed by (2) *navigating restrictive parenting*; (3) *reciprocal transnational networks*; (4) and *the becoming of women in globally interconnected settings.*

### 4.1. Navigating Bonding with Peer Groups

One of the group discussions quickly progressed after a participant burst out: 


*You lose everything if you lose your smartphone*
*!*


She explained her heart-felt experience of losing hers for 1 week. The others joined in with similar experiences and reflected on their dependency on digital access:

*You are easily forgotten if you are not on Facebook or social media. You will not receive as many invitations to potlucks, dinners, cafes, or parties if you are not on social media*. 

This view reflects a common concern of the general youth population, fear of missing out [21,22]. The peer groups of the young women consisted of (former) classmates, friends, and acquaintances in their physical and digital surroundings inside and outside diasporic networks and included youth from ethnic minority and majority groups, which makes the sample highly heterogeneous. Their navigation to avoid missing out occurred in settings where they were aware of a diversity of social norms. The need to be visible in digital settings to be included in social events in analog settings is a recurrent topic in studies on digital social youth activities, e.g., [34,43]. However, these young women also shared concerns of being looked down on by their ethnic majority peers. Nasreen describes this as follows: 


*My Norwegian former classmates are too much ‘up there*
*’. If I post something, I always feel they will find it weird. They are really picky. I follow some of them, and they follow me, but I know they don’t give me any likes. Or sometimes they accidentally like my post, and then they unlike it right away.*


Their fear of missing out took specific forms, such as struggling to belong within the general youth population and experiencing that their efforts to bond with ethnic Norwegian schoolmates on digital platforms did not necessarily open doors for further analog interaction: 


*Some people are good when they communicate with you online, but not when you meet. Then they pretend they don’t know you. It happened to me. One girl commented on my photos and chatted with me. She wrote “you are nice” and so on. Then I said “hi” to her at school and she just passed by me.*


Fears of being ignored in analog settings following digital interactions were mainly discussed in the context of being recognized by ethnic majority youth. A related worry was being viewed as friendless, particularly in arenas where they could be seen by youth from majority groups. Some of the young women shared that they support one another by clustering and by posing together to cover their vulnerability. As Shaneen shared: 


*We hug each other in photos all the time, to help hide each other (…). It is much better to be seen together than posing alone, showing yourself together with real people.*


Mutual support involved keeping track of new posts from friends and quickly posting “likes”. As Alma expressed: 


*My friends, when they post something, they call me and tell me; ‘please, like me! Support me!’. You see? It happens every day!*


Although they crafted digital visibility to avoid missing out on social events and to appear sociable and approachable, fears of being ‘tagged’ in photos complicated their navigations. For example, Fatima shared the following:


*It is sometimes hard to keep track of various snaps, because of nicknames. I don’t wear a hijab at home, and I just sent, like [a] “good morning!” [photo] to the wrong person! A boy, who screen-shot it and forwarded it to all the boys in a whole group: ’Do you want to see Fatima’s hair?’ He is hayawaan [an animal, in Arabic].*


The narrative of boys in her minority peer group seeing and sharing the photo showing her hair following her sharing mishap implies an invasion of her privacy. A fear of photo tagging is common among youth in general. However, this study infers that the consequences of being unable to manage the flow of information between digital rooms could be more severe for women, especially minoritized women. The participants discussed their fear of tagged pictures leaking to their family or diaspora network who would not share or understand the social norms of their peer group in the given context. Moreover, the young women were concerned that others could share information about them without understanding the possible consequences. They mentioned limiting contact with peers in analog settings to avoid risks of unwanted exposure. Some participants said they opted to engage only in semi-hidden private chat groups in digital spaces. Cautiousness about sharing and participating in groups thus influenced their participation both in digital and analog arenas.

Conflicting norms and moralities in their peer groups versus their other social networks spurred strategies on how to be present on SNS. On the one hand, they viewed access to different environments as an opportunity to learn about cultural differences and to collect information on various topics. At the same time, the young women tried to exhibit ‘correct’ demeanors out of concerns relating to restrictive parents and proximal family networks, which will be discussed in subsequent sections.

### 4.2. Navigating Restrictive Parenting

Conflicting norms were a recurring topic, and the young women frequently commented on their parents’ values and restrictions that limited their participation in digital arenas. They compared their parents’ restrictions and commented on how their parents’ expectations differed from those of the parents of the majority youth. In the conversations between the young women, they tended to explain restrictive parenting related to what they referred to as a “lack of understanding” of the local norms and particularly norms related to SNS. It was repeatedly stated that “They do not understand”, referring to the parents. We did not ask directly about the parents’ use of SNS or the educational level of the parents, but it must be considered that the refugee-related status of the parents implies that it is likely that they have a low educational level [44]. The differences in norms are not surprising but have other implications related to SNS participation.

One participant surprised the others by sharing that her mother sent her ‘*snaps’*. She seemed aware that this scenario would be unusual and explained with a smile that her mother shared little moments through *snaps* daily. This spurred the others in the group to share experiences and discuss how they could navigate restrictions set by their parents related to their use of SNS. How to navigate expectations from their parents while participating in online settings was a topic that led to animated discussions.

Examples, such as the one of the mother who sent snaps, also led the women to contrast their parents’ digital skills with those of the parents of ethnic majority youth. Some participants had taught their parents about SNS, for instance, by setting up user accounts and showing them how to use digital technology to connect with others. They had searched for and translated information for their parents, including health- and welfare-related information, they had helped them connect with diasporic and transnational networks by assisting them to start participating in digital networks, and some had set up digital groups for relatives across countries. This could, however, also cause friction.

As found in a previous quantitative study of youth in upper secondary schools [45] some participants experienced that they were not allowed by their parents to use SNS, such as Atifa: 

*My dad is against social media. He is so strict, and he wants to know what’s going on. He thinks about reputation. You know, like she said, I don’t think it’s ok. If I was a parent, I would say let them [the parent’s children] try, to learn. He doesn’t know how to use social media, but he still has very strong opinions. It’s about culture. Some Somalis think that because we are Somali, we should not post anything about ourselves on social media. We should not be visible out there*.

Although the young women taught their parents about SNS, they experienced that their parents were cautious or negative about their daughters’ use of SNS. Like in the citation from Atifa, the restrictions from parents were frequently linked to what the participants experienced as culture-specific norms, which in most cases were norms that were specific for women. The participants also recognized the low e-literacy skills of their parents as an advantage because it implied that the parents would be unlikely to observe how they used SNS or to use spyware or location tracking to follow the movements and social relations of their daughters. However, others stated that parents with low e-literacy collaborated with other parents to monitor the activities of their daughters through SNS. The young women limited their parents’ access to information by posting less about themselves, by using different profiles and platforms, in particular platforms that allowed anonymity or options for limited sharing of personal information, and by digitally connecting with fewer people, especially persons who might share information with their parents or who might not understand their situation. Several participants mentioned platforms where they could ask questions and participate without revealing their identity, and they considered these to be important. 

Digital communication could also cause misunderstandings with parents due to a lack of context or language barriers, as illustrated by Amira: 


*The translation function on Facebook is not the best one. Once my parents saw that I had called someone my ‘girlfriend’*
*, and it led to a big talk where I tried to explain that we were not lovers. It was very difficult. I try to be more conscious, but it’s kind of fun that there are so many things that can lead to misunderstandings.*


Although the women mostly referred to experiences that portrayed the parents as restrictive, there were exemptions. A few parents supported their daughters’ endeavors, for instance, by refraining from responding to extended family networks who attempted to gain information about the young women via digital platforms. Some parents had rejected inquiries from relatives about their daughters on digital platforms. As Sara shared: 

*Once when I did not answer [via SNS platform], they called my father. They [distant relatives] complained, but he told them that I had to do sports. My parents are kind*.

Thus, even if the digital presence of parents limited the SNS activities of the young women in relation to their peers, the supportive approach of a few parents also served as a safeguard against interference from diasporic, extended family and from transnational networks more broadly. 

### 4.3. Navigating Reciprocal Transnational Networks

The young women maintained contact with diasporic and transnational networks. Diaspora refers to religious or national groups living outside a (imagined or ancestral) homeland [46], whereas a transnational network is a broader term, which includes all forms of social formation, social networks, groups, and organizations related to a (imagined or ancestral) homeland. The participants described how the networks exchanged resources, including information, and could be mobilized. This phenomenon has been referred to as reciprocity, understood as a resource exchange, or ’gift-giving‘, that develops and sustains social connections [47], almost as a strategic way to access resources. The use of reciprocity in diasporas has been documented in previous research, and the term has been refined [48,49], including reciprocity in relation to SNS participation [49]. The participants described how SNS add to the dynamics of these reciprocal networks because they enable direct, immediate, and continued contact. Aisha narrated the following: 

*In the group chat, we can see how people are doing, or when anyone is ill, has died, or gets married. Sometimes they share the word of God with the whole family. We have different family groups, and I am in five. We have various groups for practical help, like collecting money to help someone. Not everyone joins, but the coordinator sends invitations. We are just a normal family. Immigrant families need digital technology to do the things all families do. Everyone I know has such groups*.

The participants valued reciprocity, including a sense of trust that the groups could mobilize, which promoted a sense of belonging and continuity, as also observed by previous research on diasporic networks (Phillimore). Several participants had themselves initiated groups that included older generations because they wanted to contribute to upholding networks across distances. 

Other participants sought support to cope with childhood experiences and to address emotional strains during their transition to adulthood, as put by Daneen:

*It is the first time some of us that are young stay in contact and try to have relations with those who are in (home country), the others who are here, and all those that we know from the year we lived in (another country). I was only a child, but now I try to communicate with those who were there. It’s important to stay in touch so that I can ask: Where were you when I was a child and I needed you? Now I am almost an adult, and I must start over. It’s important to talk about the things that happened*.

The discussions of the young women shifted towards challenges, which frequently revolved around a lack of control over information flows across various groups, leading to challenges in maintaining privacy. This phenomenon is a general trait of SNS and is referred to as context collapse [43] but was associated with more serious challenges within the participants’ diverse networks. The young women discussed strategies for coping with participation in digital networks with different norms, as in this conversation between Amiina, Fatimah, and Salma: 


*Even if you change only a little, they [family in the place of origin] think you changed totally. In Norway, it’s just normal to go for walks in the mountains, even in the rain. But when the family see a photo like that, they start talking, because they do not understand it. So, there’s pressure both on social media and different channels. My friend said to me: ‘You have a lot of money. You are always in nature*
*.’ I said I don’t have a lot of money. She said: ‘You go on walks to take pictures*
*.’ Yeah, they think a lot.*


*Yes, I’ve experienced a bit of that too. One of the relatives I got to know recently on Facebook told me ’You can’t speak your language. You don’t know much about your culture anymore. You’ve changed completely. You are not in contact with all your relatives in Somalia.’ But I do have a daily life here where I go to school, and I have things to do. I can’t call all my relatives all the time and ask if they are doing well. Also, I can’t make sure they’re doing well. And yes, I must learn Norwegian, but I’m still Somali and I can still speak the language. My friend doesn’t believe that I’m Somali anymore. He says you think you’re Norwegian. He referred to my photos, and where I had been. He said that I believe that I am rich now. That’s how it is. They see photos in their own way. Social media leads to a lot of misunderstandings. I try to use it less now. It just leads to stress and conflicts, because they don’t understand*. (Fatima)

*If they had known all the social media we use, they would have thought that we were total highflyers. It is not normal in their culture to have social media where you post images. They are just following us to see what we are doing. You just have to keep some distance, because they don’t understand that the culture we have here in Norway is different and that sharing images is completely normal*. (Salma)

The young women used words such as “pressure,” “stress,” and “conflicts” to describe what they oftentimes consider “misunderstandings” due to the limited knowledge of the diasporic and transnational networks about the youth culture in Norway. A recurring topic was that they experienced accusations of forgetting and failing their people, language, culture, and customs because they had become *foreigners* or because they view themselves as *better than others*. A familiar complaint was being called “rich” and “successful” and accused of not caring about relatives or diaspora members. These comments influenced their strategies for digital participation, but it was not explored how or whether these comments influenced their participation in analog settings. Although the young women expressed that the comments led to stress, they valued these networks and, thus, aimed to balance diasporic norms against those of peer groups. Their main strategy for managing divergent norms involved sharing less text and fewer images, displaying only parts of their face through artful techniques, or replacing profile pictures with meaningful images such as a saying, poem, or sports club logo. They found it challenging to show older distant relatives respect by avoiding breaking norms. They experienced negative comments and behavior that they found difficult to cope with, for instance, publicly posted negative comments on their clothes. Still, some women mentioned that they did not see it as an option to block them from commenting. Moreover, as the young women discussed their strategies to avoid hurtful situations involving their diasporic connections, they simultaneously feared being too invisible, forgotten, or difficult to be found on SNS by their peers:


*People need to recognize me, so I have a profile photo that I like. Only my face, not everything else. And I have the correct info, my name, and I never change it. Or perhaps in five years.*



*Some people only show half of their face, but I think that you should show your whole face and your last name, because then people can find you, and start following you. Otherwise, you may lose friends, who are searching to find you for years.*


Although their attempt to manage information flow is a general trait of SNS behavior [34], their fear of sanctions severely limits their digital participation compared with their majority peers. Moreover, they were aware that SNS implied that they potentially could be seen in any situation at any time, as others could also share information.

### 4.4. The Becoming of Women in Globally Interconnected Settings

The young women expressed sadness and oftentimes anger when sharing experiences related to negative feedback and negative social control. They found difficulty in avoiding failure and coping with gender-specific expectations from their extended networks regarding their presence on SNS.

*We should be private. We should just stay within the family. It’s because people may gossip if they know anything about you, and then things can get out of proportion. For instance, if your clothes are too tight, it means that you have been naked and that you have been naked with boys. So, being present on social media means that you can just be seen as doing something bad*.(Jasmin)

The participants discussed gossip and judgmental comments about their appearance. Moreover, negative judgments and the awareness of potential exclusion from diasporic and transnational networks were part of their experience of becoming women.


*They gossip in a very judgmental way. Even if you do not show any of your skin, but only pose in a way that does not reveal anything, they still judge. They judge you no matter what you do. The worst is when they stop talking to you but only talk to your family, so that they judge you too. It is very annoying how they interpret things all the time. They do not care about you.*
(Sadia)

In this group discussion, a participant narrates how relatives told her that she had forgotten her culture because she did not cover her hair and did not wear traditional clothes.

*Once I put up a picture where my hair was fresh and curly. Suddenly I was famous. They didn’t like it and had a lot of comments. They called me. That’s why I’ve stopped using pictures. It’s a bit difficult when many people think that those who are in Europe forgot their culture, but I never did because I grew up there and my culture is in my head. In Eritrea, I had to wear a hijab and I had such a dress, so they know me like that. I’m not Muslim, but I like my hijab*. (Selma)

Others were told to act in line with their culture or to follow their religion. As narrated by Aiisha: 

*My aunt and cousins called me up and told me to stay more in control and to follow my religion. Multicultural women get a lot of these comments*. 

Despite such comments, none of the participants considered open resistance to critiques. As cited in previous excerpts, the young women experienced scrutiny of photos that displayed their hair, face, clothing, skin, or body. Even those who typically do not cover their hair were prompted to change their digital self-presentation due to negative feedback. Others experienced accidental hair exposure despite their efforts to avoid this, such as tagged photos. A few experienced blackmail and threats for exposing their hair. In response, the participants employed various strategies to navigate SNS and protect their privacy. Others had abandoned SNS platforms to avoid stress from having to relate to conflicting norms across networks but returned to access information from peer networks, from school, and about extracurricular activities. Another strategy was being a passive member in groups on SNS in order to avoid sharing information that may be passed on by others. Others commented that this strategy implied that they were not, in fact, participating. Yet another strategy was reducing their visibility, in order to appear modest and humble, such as by refraining from posting photos or by refraining from posting outside private groups. Some participants displayed a thorough knowledge of options provided by different apps, and they tailored their activities accordingly.

*I change my profile picture only every six months on Facebook, because I have older relatives there. But on WhatsApp [I change it] every three days. On Facebook you must put the picture up so people can like it, and therefore it gets more attention. On Insta you can swap it out without people noticing, because then it’s not as a post*. (Alma)

Knowledge about the technical options of the various SNS platforms provided opportunities to avoid visibly breaking gendered norms for behavior in diasporic and transnational networks. 

The participants also discussed how they related to men who were part of their diasporic or transnational networks, or who pretended to be, in digital settings. As Ilham puts it: 


*We receive a lot of those messages from creepy psycho-dudes who see that we live abroad. They come from Iraq, or I don’t know where. You block them and they just create a new user!*


Although Norwegian women are more susceptible to encountering negative experiences on SNS than Norwegian men [31], strict norms that limit interaction with men to preserve *honor* add to the complexity of the platforms for the young immigrant women in Norway who participated in the study described in this paper. 

Simultaneously, the option to connect with diasporic and transnational networks in digital settings could widen the range or knowledge of potential partners within these networks. However, even for those who find a partner via SNS, technological options can create new challenges: 


*“My cousin got married after meeting her husband via Facebook. They talked for a year or two, got to know each other a little, and then they got married. Another friend got married too, after three years of video talks. But they used filters when they talked. She used a filter that changed her body. He came to Norway after three years, but then he said: This is not you! He broke up with her, but it was her fault because she used too many filters.”*


The other participants commented that both men and women can deceive each other in digital settings in ways that represent a breach of trust.

Diverging expectations related to gender and negative social control tendencies were challenging for the participants, although they simultaneously valued belonging to varied local and transnational networks and they rarely confronted condemning remarks. While the group was heterogeneous in many ways, the women found it enlightening to share experiences, and they suggested that this practice should continue for the benefit of others struggling with similar challenges.

## 5. Discussion

Contextual factors linked to social, cultural, and economic aspects of social class status influence the relationship between individuals and their participation in different arenas, including SNS. However, less is known about SNS participation, the dynamics and mechanisms thereof [10,25], and how this influences health and wellbeing [23]. 

The young women who participated in this study described their navigations and contexts on SNS in ways that illuminate their challenges and resources regarding participation. They participated in heterogeneous diaspora networks and strived to connect with ethnic majority peers through SNS. Similar to ethnic majority youth [14,42], they experienced that SNS could facilitate social participation and connectedness and that digital visibility could enhance their social participation also in analog settings. They reported that fear of missing out could lead to stress, pressure, and conflict, which should be further elucidated in light of their positions within and their cautions about their diasporic and peer networks, which exhibit different and frequently conflicting norms. Their experiences of efforts to connect with ethnic majority youth and to understand those youth’s cultural codes while simultaneously endeavoring to avoid negative sanctions from their diasporic networks resulted in strategies where they carefully managed their visibility and social participation. The balance of being visible and staying within the norms of the diaspora led to the fear of missing out, and actually missing out, in specific ways. They were aware of the negative consequences of being less visible but perceived this as a necessary decision. Previous research found that restrictive parenting strategies include surveillance of SNS [50]. The prevalence of parental control on SNS appears to be similar across groups [51], but the consequences of such control differ [52]. The participants reported that they limited their participation in digital settings such as SNS due to conflicting cultural norms, although some expressed that their parents intended to support them to a certain extent vis-à-vis more restrictive members of their diasporic digital networks. The kind of limitations they experience should be explored further, as passive use of SNS is strongly associated with negative consequences for mental health and wellbeing [23]. 

The consequences of their orientation toward divergent social networks are also evident in the way they compare their own situation with that of both their majority peers and peers in diaspora and transnational networks who live in different and, frequently, more challenging circumstances. While ethnic majority youth in the Nordic countries tend to compare themselves to other ethnic majority youth who they perceive to be in a *better* social position than themselves, the participants in this study also compared themselves to peers in other situations. The tendency to compare to peers in situations perceived to be better positioned has been hypothesized to reduce wellbeing [13]. The orientation toward divergent networks appears to involve efforts to understand and relate to various cultural codes and situations. The effect of this tendency on wellbeing remains unknown. 

The current study illustrates the relevance of theory on youth development, emphasizing how interaction with peers comprises an important source of learning about norms in complex societies [5,32]. This complexity increases as the participants increasingly relate to social networks across cultures, geographical distances, and intersecting digital/analog settings. Participation in digital and analog settings was clearly interlinked. For example, the possibility of being tagged implied that information about activities in analog settings could reach unintended audiences. The study found that the participants frequently coped with risks of experiencing context collapse [34] and conflicting norms by evading interactions with peers and by limiting activities with friends in both digital and analog settings. 

Young women face greater challenges than men [13,31], and our findings show that young immigrant women face additional challenges. Some challenges are seemingly related to what has been referred to as an *honor code* and the transition into honorable women, e.g., [50,52]. The idea that the honor of young women and that of their families is at stake [52,53] also exists in ethnic majority populations in northern Europe [52,54]. However, the limitations placed on their participation are less extensive; expressions of condemnation are subtle; and the consequences of failing to observe the code are less grave [45,55]. Moreover, less discrepancy occurs in norms between generations [55]. Additionally, women experience more gender-related hostility [13,18] and tend to be more concerned about, and tasked with, maintaining social harmony on SNS than men [31]. 

Contrary to previous research that highlighted the potential of social participation on SNS [8], specifically as platforms where youth with immigrant backgrounds develop cultural identities and counter exclusion [20,28], the participants tended to limit SNS participation. Furthermore, their presence on SNS tended to reinforce their segregation because they limit their participation in physical arenas since information about their participation could be spread via SNS. Other studies also reported processes of exclusion and reinforced segregation and differences between groups [19,25]. 

It seems likely that the participants’ caution about and occasional avoidance of participation may have implications for the development of participatory skills, in line with the typology of youth participation suggested in [4], where the authors argue that young people may need supportive environments to develop a sense of empowerment and social engagement. The development of skills for *social participation* prior to skills for participation as *social engagement* has been suggested in previous studies but is partly countered in recent research [5,20]. This should be investigated further while also considering that other kinds of participatory skills are potentially learned by participating in diasporic networks and by relating to and navigating divergent norm systems and cultural codes. 

The current study found that SNS are settings for inclusion and exclusion processes, as the young women cautiously explored social participation beyond their diaspora networks, in line with studies of young immigrant women with a Muslim background in Denmark [26,35]. However, they clearly experienced specific challenges due to their positions within and across diverse networks that coexist in partially overlapping SNS settings and networks. These diverse settings simultaneously provided arenas to learn about different cultural norms, which in varying and sometimes conflicting ways define *correct* uses of SNS and inherent context collapse. The young women expressed that the exchanges of experiences about strategies for navigating social networks that took place in the dialogue groups organized for this study were helpful, and they expected that similar exchanges of experiences could be useful for others too. Their willingness to share their experiences with the researchers and with each other must be seen in light of their participation in the NGOs. Their participation in the NGOs involved social participation and exposure to conversations about topics such as negative social control. This may have made them more likely to be aware of and able to share experiences related to this and other relevant phenomena. Their participation in the organizations implies that they are women who are active in an organized, semipublic arena associated with the aims of motivating social participation and social engagement in young women with migrant backgrounds. Further studies could explore specific segments and their participation in specific SNS.

Previously, scholars have called for further studies on SNS uses by specific groups to inform the development of health promotion strategies [9,17]. Such studies are still scarce, and few interventions are adapted to specific target groups by employing knowledge about conditions in their specific SNS settings [3]. Based on the results, this study emphasizes three characteristics to be considered.

First, strategies for health promotion must acknowledge that digital and analog settings intersect. Thus, the social contexts of targeted groups must be carefully considered. As is evident in this study, the young women with ethnic minority backgrounds were acutely aware of conflicting norms, of risks of context collapse [34], and of exclusion and self-exclusion as a result of information shared on SNS. Expectations from heterogeneous networks may consequently limit their workable or acceptable strategies for participating in activities with peers. Whereas SNS are important social settings for promoting participation and countering exclusion and where personal skills can be developed for all groups, SNS must also be recognized as potential platforms where inclusion *and* exclusion processes can be reinforced, as is the case for analog social settings where youths interact. Additionally, the study highlights that anonymous chat platforms are considered useful by the target group for discussing issues that they were unable to discuss with their parents or openly with others, including issues related to negative social control. 

Second, in addition to using SNS to search for health-related information, the participants also assisted relatives with low levels of e-literacy by sharing information and establishing groups for communication within wider social networks. This tendency may support earlier suggestions for health promotion by amplifying selected voices on existing SNS instead of setting up information sites [12], in addition to tailoring information to diaspora networks, with attention to social media sharing. The recent pandemic illustrated the importance of disseminating health information within minority groups (e.g., [56]). Strategies used to reach others with limited participation in public platforms require further examination.

Third, the cocreation of health promotion strategies should involve young immigrant women and their communities in digital settings because their digital and analog environments are constantly changing [57]. Through sharing experiences in focus groups, the young women began exploring the limitations of and possibilities for various forms of participation. This process may be further developed. Thus far, however, professionals mainly develop health-promoting platforms on SNS without coproduction with young people or other user groups [12]. Participation of target groups in developing designs and strategies for health promotion is rare [3]. It has been suggested that professionals should be involved as supportive outsiders [12] who strengthen youth voices or invite them to develop strategies. Media educators suggest that the active production of media materials can lead to youth empowerment through the development of self-expression, youth voice, and agency, e.g., [58]. Strengthening their voices and promoting participation are imperative for groups at risk of becoming muted in digital settings such as SNS because such platforms have emerged as important settings for interaction and learning among adolescents. Active voices and counternarratives can be constructed and expressed through participation and coproduction in order to address issues and develop possibilities related to situations where they learn, work, play, and love.

## 6. Conclusions

The study shows that young women with immigrant backgrounds encounter specific challenges and resources related to participation in digital settings such as SNS because they participate in networks with various norms, knowledge, and resources. Their strategies in digital settings reflect and mutually influence their positions and dilemmas in analog settings. Tendencies toward exclusion, segregation, and negative social control were reproduced in digital settings but were also contested through attempts to establish semiprivate arenas within the settings. They feared the consequences of context collapse on SNS and consequently tended to limit participation in both digital and analog settings. Notably, the ability of SNS to bring their local and global networks together can actualize resources and conflicts simultaneously, presenting a paradox.

The dynamics of their participation have implications for the design of health promotion strategies and illustrate the importance of tailoring interventions to specific groups and contexts. First, strategies for health promotion must acknowledge that digital and analog settings intersect. Thus, the contexts of groups must be considered when developing health promotion strategies. The young women expressed the need to learn about local norms for SNS participation and for anonymous chat services where they can learn about topics that their parents consider taboo. Second, they expressed that they help others by sharing information and setting up groups for diaspora and transnational networks. This may constitute a channel for disseminating information to their networks. Third, the cocreation of health promotion strategies can and should be used to tailor interventions, to strengthen voices, and to promote participation for groups that are at risk of becoming muted in SNS arenas. It is promising that the young women who participated in this study would like to contribute to improvements for other young women by cocreating strategies for health promotion in digital settings.

## Figures and Tables

**Table 1 ijerph-20-04033-t001:** Example of structured evolution of themes.

Transcript *An extract from a section of one focus group. Several extracts were generated across the three focus groups, and these constituted a basis for suggestions for discourses* via *an open thematic coding phase*	Open coding *Three guiding principles: repetition,* *recurrency, and forcefulness*	Discourses based on the open coding phase	Closed coding *Guiding exercise: link the open coding discourses with public health perspectives by asking critical questions regarding the material*	Adjusted themes Guiding exercise: advance the analysis by considering selected perspectives on agency and participation
*Some people are good when they communicate with you online, but not when you meet. Then they pretend they don’t know you. It happened to me. One girl commented on my photos and chatted with me. She wrote “you are nice” and so on. Then I said “hi” to her at school and she just passed by me.*	Striving to belong Fears of being ignored	Power relations and status-based hierarchies in youth groups	How may power relations and status-based hierarchies in youth groups link with public health ideologies in the context of the subjects and their navigations on SNS? Perspectives on agency and participation for young migrant women.	Navigating bonding with peer groups

**Table 2 ijerph-20-04033-t002:** Overview of the four themes.

1. Navigating bonding with peer groups 2. Navigating restrictive parenting 3. Navigating reciprocal transnational networks 4. The becoming of women in globally interconnected settings

## Data Availability

Data are available on request.

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
