# Peer review of "Developing Public Health Promotion Strategies for Social Networking Sites: Perspectives of Young Immigrant Women in Norway"

_ijerph, 2023, doi:10.3390/ijerph20054033_

Round 1
Reviewer 1 Report
There is no theoretical framework in this study. Throughout the manuscript, the theoretical contribution is not completely clear. More importantly, the finding and discussion sections are primarily descriptive about the women’s use of social media. It is hard to tell the relevance of such use to health condition. The readers are hard to find meanings (theoretic contributions) beyond empirical data. It was unclear what the study was contributing from a theoretical perspective to health promotion.
In this paper, different social media are treated as a whole while in fact, their impacts on health condition could be very different. The overgeneralization greatly confined the significance and novelty of the findings.
The limited representativeness of the sample, only 15 participants, may greatly weaken the strength of the findings.
This issue is perhaps also compounded owing to the way that the interview data is used in the article, as we often know little about each woman other than simply their fictive names and their interview quotes. I’m left wondering whether (and if so, how) factors such class, education, literacy, and wealth might have influenced their experience of using social media, and more importantly, to what extend the findings could be reproduced on other groups from different backgrounds. Thus this deserves more critical discussion.
Coding details should be provided in the manuscript, more information should be provided. Supposedly, the authors should explain how the themes were developed.
Author Response
Dear reviewer,
thank you for your constructive comments and thorough reading of our manuscript. We found the comments very useful for the revision of the article.
See the attached overview of our responses point-by-point.
Best wishes.

Reviewer 2 Report
Dear colleagues,
I am very grateful to the authors of this important and engaging applied research. A huge qualitative database and a great contribution to understanding the working side of online health services, their inherent problem and ways to ameliorate them.
The paper is excellent, and my pieces of advice are aimed at making the best use of your database and making your argument even more readable and impactful for both the scientific community and practitioners in health promotion.
The first point to which I want your attention is that it would be very useful to add in appendix your qualitative questionnaire and list of questions.
Also, your statistical section is not enough: you need to add the regression model, as well as complete figures.
Concerning the discussion section: I want you to stress more the fact that your results could help to improve our knowledge of internal validity (and please be precise on how) and external validity (same: please be precise).
The fundamental dimension of online advice may also be related to issues of trust between youth women and health promotion workers, which may be influenced by cultural and ethnic issues. You quote the research findings of Vacca et al. published in the Journal of Ethnic and Migration Studies (2022; 48:13, 3113-3141, DOI: 10.1080/1369183X.2021.1903305), Beyond ethnic solidarity: the diversity and specialisation of social ties in a stigmatised migrant minority.
Line 67, there is a study realised by Morelli and published in Social Science Information, 59(4), 679–703. https://doi.org/10.1177/0539018420973596: A study of Italian youth’s self-managed dairy restriction and healthy consumerism. ît shows when youth use online health websites/blog/chats and when they go to their general practitioner.
Line 87: social participation is not a clear concept. what do you refer to?
Line 101, please explain better why you say exclusion here, and then justify the use of a positive focus on inclusion.
Line 163 and 285, could you please define reciprocity? For example, Pais and Provasi (2015), employing the Polanyian forms in investigating the sharing economy, propose three variants: Collaboration, Reciprocity in the strict sense, and Common- pool arrangements, while Sahlins (1972) makes a distinction between Generalized, Balanced and Negative reciprocity. Being so important for your paper, more than reciprocity is needed to account for the full complexity of emerging trust or for breaching of trust (line 426). You can break down reciprocity into three further forms so to say Instrumental reciprocity, Mutualistic reciprocity and Communitarian reciprocity, as done by Giorgi, A., (2017). Migrants in the public discourse: Between media, policy and public opinion. In Trade unions and migrant workers. Edward Elgar Publishing.
Line 442 - if FOMO could lead to stress, please anticipate it in the results, and then discuss it extensively here in the section 4.
Line 534, concerning self-expression: a key point you emphasise is the significance of the findings you have accurately documented. In particular, the point about the importance of assessing the youth's understanding of the advice received. This is a question of communication style but also of protocol: it is necessary that even under very tight time constraints, health promotion workers can always summarise their advice and make sure that there is understanding by youth women. This is an organisational innovation that can easily be extended to other territories beyond Norway. This organisational innovation can be consolidated by generating real institutional learning in the local regulatory system and in the protocols concerning the modalities of online screening and help conversation. In doing so, it can have a transferability and replicability potential and become not a protocol by a communication style that can easily be extended to other territories. It does, however, require that health promotion workers learn how to check that parents have understood. And they learn how to do this even with low-educated parents as shown by Cousin et al. in Journal of Ethnic and Migration Studies 47.13 (2021): 2938-2960 insisting on the possibility of organising rapid training in this regard, with short modules to simplify language and verification.
There is probably a point related to trust and distrust that could be better developed. Probably you can mobilise classic work by Weber and Schluchter. Trust conceptualized in a Weberian manner, is strictly related to conventional forms of link between individuals, procedures, and institutions. Trust in migrants and their family-making practices is not structured by impersonality or regulation, but by the lacking of proof of recognition, as developed by Morelli in Social Sciences Information.
Overall, this is an important subject and could become an excellent article, but the current version would benefit some more clarity and much elaboration in dealing with the powerful linkage between the theoretical and the empirical parts of the research.
Overall, I thought this is an important, deep, and generative article and I hope the above comments are helpful to the authors.
I hope to see your contribution published as soon as possible.
Author Response

(The authors gave the same response as above.)

Round 2
Reviewer 1 Report
I applaud the authors on their efforts to improve this manuscript. I found the addition of information on research design and coding process to be useful.
However, I do think findings section can be enhanced and would like to ask the authors to strengthen their analysis so as to contextualise the participants’ experiences a bit more.
For instance, in theme 2 - Navigating restrictive parenting, the analysis mainly focused on the parents’ low e-literacy skills, and the findings are generally common in relevant studies on parenting relations and SNS use. I encourage the authors to bring in more in-depth review and discussion about the uniqueness of this topic, esp. in relation with the participants’ ethnic minority or refugee statuses.
Similarly, for theme 4 - Gender-specific expectations, the authors are expected to further elaborate the uniqueness of the participants’ experience when they try to navigate two independent social networks. Importantly, the discussion should be situated within the contextualized meanings of two contrasting cultures.
Also, I would recommend deleting unnecessary quotes. For example, quotes like the one in line 373 and 374 does not provide much nuance or specifics and should be deleted in my opinion.
Author Response
Thank you for your constructive comments! See the overview over our responsens in the table below, and the changes in the documents marked with yellow.
I applaud the authors on their efforts to improve this manuscript. I found the addition of information on research design and coding process to be useful. |
We are pleased that you found the improvements and the added information about design and coding useful.
|
The findings section can be enhanced and would like to ask the authors to strengthen their analysis so as to contextualise the participants’ experiences a bit more. |
We have made changes throughout the findings section, by adding information about context. See text marked with yellow. |
For instance, in theme 2 - Navigating restrictive parenting, the analysis mainly focused on the parents’ low e-literacy skills, and the findings are generally common in relevant studies on parenting relations and SNS use. I encourage the authors to bring in more in-depth review and discussion about the uniqueness of this topic, esp. in relation with the participants’ ethnic minority or refugee statuses. |
Thank you for this suggestion. The additions and changes are marked with yellow. We believe this have improved the text. |
Similarly, for theme 4 - Gender-specific expectations, the authors are expected to further elaborate the uniqueness of the participants’ experience when they try to navigate two independent social networks. Importantly, the discussion should be situated within the contextualized meanings of two contrasting cultures. |
Yes, this is an important finding that we now have aimed to highlight. |
I would recommend deleting unnecessary quotes. For example, quotes like the one in line 373 and 374 do not provide much nuance or specifics and should be deleted in my opinion. |
We believe that quotes can be illustrative, and have added some context to the quote that you mention. The specific quote has been included to illustrate nuances and this has been complemented by adding a sentence in the main text. |